# Advances and Challenges in Diagnostics of Toxoplasmosis in HIV-Infected Patients

**DOI:** 10.3390/pathogens12010110

**Published:** 2023-01-09

**Authors:** Roland Wesołowski, Marta Pawłowska, Małgorzata Smoguła, Karolina Szewczyk-Golec

**Affiliations:** Department of Medical Biology and Biochemistry, Ludwik Rydygier Collegium Medicum in Bydgoszcz, Nicolaus Copernicus University in Toruń, 87-100 Toruń, Poland

**Keywords:** toxoplasmosis, HIV, diagnostics, *Toxoplasma gondii*

## Abstract

*Toxoplasma gondii* is a worldwide distributed protozoan parasite. This apicomplexan parasite infects one-third of the population worldwide, causing toxoplasmosis, considered one of the neglected parasitic infections. In healthy humans, most infections are asymptomatic. However, in immunocompromised patients, the course of the disease can be life-threatening. Human immunodeficiency virus (HIV)-infected patients have a very high burden of *Toxoplasma gondii* co-infection. Thus, it is essential to use modern, sensitive, and specific methods to properly monitor the course of toxoplasmosis in immunodeficient patients.

## 1. Introduction

*Toxoplasma gondii* is an obligate intracellular protozoan parasite of 300 species of mammals and more than 30 species of birds [1]. The parasite occurs worldwide, infecting one-third of the population worldwide [2,3,4]. *T. gondii* is a pathogen of relevant medical and veterinary significance [3]. It is a unique parasite whose interesting development cycle makes it possible to infect humans in various ways, mainly by consuming raw meat containing tissue cysts and accidentally ingesting oocysts in food, water, or soil contaminated with cat feces. The parasite can also penetrate through the placenta from an infected mother, by organ transplantation, or by blood transfusion [1,4,5]. Pinto-Ferreira et al. [6] noticed that transmission routes of *T. gondii* varied by decade. In the 1960s and 1990s, the ingestion of cysts in meat and meat derivatives was the primary source of the parasite. In the 1980s, outbreaks occurred mainly through milk contaminated with tachyzoites. Interestingly, in the 2000s, the parasite spread among humans mainly through oocysts found in water, sand, and soil, while in the 2010s its spread was mainly due to oocysts present in raw fruits and vegetables [6]. The authors noticed that over the years meat became safer due to improved management and reduced pathogenic animal infections as a result of the development of livestock farming technology. They also suggested that, in the past 20 years, consumption of healthy types of food, such as vegetables, increased (among others, through efforts to introduce healthy eating habits to combat obesity and other metabolic diseases). Toxoplasmosis infections related to consuming vegetables occurred mainly as a result of the contamination during production, but also during processing and consumption [6]. Due to their scavenging eating habits, free-range chickens play a substantial role in the epidemiology of toxoplasmosis [7]. Therefore, chicken meat appears to be a significant source of infection for both cats and humans. Balbino et al. [8] suggested that fruits and vegetables contaminated with oocysts were the most described transmission routes in outbreaks of toxoplasmosis in Brazil. It seems that contact with cats can no longer be considered a significant risk factor for *Toxoplasma gondii* infection in urban areas [9]. Recent papers suggest that there is also a possible sexual transmission of toxoplasmosis [10,11,12]. Tong et al. [11] demonstrated the presence of *T. gondii* cysts in the ejaculate. Moreover, Kaňková et al. [12] hypothesized that transmission of the parasite is possible by oral sex, particularly when infected sperm is swallowed by an uninfected person.

Even though toxoplasmosis has been known since 1908 [1], an effective vaccine for humans against this disease has not been yet developed. Nevertheless, vaccine development research is ongoing, and several vaccine candidates have been examined in different animal models, which undoubtedly brings us closer to achieving the goal. A live-attenuated vaccine based on tachyzoites of *T. gondii* with a very short shelf life of 10 days is available, but its administration is restricted to veterinary use only [13,14,15].

Although prevalence and mortality of toxoplasmosis have decreased over the past 20 years, the national public health agency of the United States, namely the Centers for Disease Control and Prevention (CDC), considers this disease a neglected parasitic infection demanding public health action [16]. Clinical symptoms of toxoplasmosis are mild and self-limiting in immunocompetent humans [1]. However, the disease’s oligosymptomatic course is characterized by malaise, lymphadenopathy, and fever. Although most toxoplasmosis cases in immunocompetent humans have been reported as asymptomatic, there are groups of risk in which the course of the infection can be symptomatic and even severe. High-risk groups for the severe course of the disease include immunocompromised patients (i.e., people suffering from acquired immunodeficiency syndrome (AIDS) or cancer), immunosuppressed organ transplant recipients, and fetuses in the case of congenital infection [17,18]. In immunocompromised patients, symptoms can be severe and associated with more complications, including fulminant myocarditis with heart failure, encephalitis, splenomegaly, retinochoroiditis, and pneumonitis [18,19,20].

Toxoplasmosis has been reported as the most common opportunistic cerebral infection in human immunodeficiency virus (HIV)-infected patients [21]. Moreover, it is considered a frequent cause of mortality amongst severely immunosuppressed HIV patients. Ocular and cerebral toxoplasmosis are fundamental problems in AIDS patients and may be considered factors suggesting AIDS infection [4]. In the meta-analysis, Wang et al. [13] indicated that HIV patients are characterized by a very high burden of *T. gondii* co-infection, with the highest percentage in sub-Saharan Africa (about 87.5%). Therefore, they highlighted the importance of routine testing for the presence of *T. gondii* in all people with HIV infection.

Since immunocompromised patients are a particular risk group, knowledge of traditional and new diagnostic techniques is crucial for the selection of an appropriate diagnostic method enabling rapid and accurate diagnosis as well as timely and effective treatment [22]. Clinical manifestation of toxoplasmosis is non-specific and often unreliable. Thus, the disease diagnosis is traditionally based on laboratory testing. Accurate and early diagnosis is pivotal for the prevention and treatment of the severe course of toxoplasmosis. This requires using specific and sensitive diagnostic tests to help detect the infection efficiently. Serological methods based on detecting anti-*Toxoplasma* antibodies are commonly used in routine diagnostics [22]. In this review, we discuss currently available data on serological and molecular methods used in the diagnosis of toxoplasmosis with emphasis on HIV-infected patients as a group at risk of severe complications of this opportunistic disease.

## 2. The Course of Toxoplasma Infection

*Toxoplasma* has a complex life cycle consisting of sexual reproduction, namely sporogony in definitive hosts (felines), and asexual reproduction, namely schizogony in intermediate hosts (warm-blooded animals, including humans) [1,23]. In intermediate hosts, parasites released during the rupture of oocysts or tissue cysts (containing bradyzoite stages) infect enterocytes and transform into fast-replicating tachyzoite stages. Then tachyzoites proliferate intracellularly inside a parasitophorous vacuole, which separates the parasite from the cytoplasm of the host cell [23,24]. The parasite primarily attacks nervous and lymphatic tissues, heart, and lungs [25]. Tachyzoites can transform into metabolically inactive bradyzoites and persist intracellularly in tissue cysts. Sporadically, tissue cysts may rupture to release bradyzoites, which can be killed in immunocompetent hosts. In immunocompromised hosts, e.g., HIV-infected patients, released bradyzoites may proliferate locally and invade other organs [1]. Bradyzoite-containing brain cysts may reconvert into cytotoxic tachyzoites and cause toxoplasmic encephalitis (TE) [24]. Toxoplasma infection may occur in patients after solid organ or hematopoietic stem cell transplantation as a result of the reactivation of pre-existing latent infection, a newly acquired food-borne infection, or the presence of tissue cysts in the transplanted organ [25].

## 3. HIV Infection and Toxoplasmosis

HIV infection is one of the major causes of morbidity and mortality worldwide, with most cases occurring in sub-Saharan Africa. Over 75 million people have been infected with HIV worldwide, and around 37 million live with this infection [26]. HIV is a retrovirus with a long latency period of slowly progressing course of the disease. Two main types of the HIV, namely HIV-1 and HIV-2, exist. HIV-1 is the prevalent and most infectious strain responsible for the global HIV/AIDS epidemic [27]. It primarily attacks CD4^+^ T cells, which allows the virus to destroy completely the immune defense of the organism [26,28]. The virus has also been revealed in macrophages in diverse anatomic locations [26]. After the infection, the HIV envelope glycoprotein gp120 binds to the CD4^+^ T-cell receptor (TCR). It then binds to a co-receptor, most frequently either CCR5 (C-C chemokine receptor type 5) or CXCR4 (C-X-C chemokine receptor type 4) [29]. In the next step, HIV reverse transcriptase produces DNA from the virus RNA and HIV integrase inserts this DNA into the host cell genome [30,31]. Once transmitted, HIV takes hold in the mucous membranes and spreads to the lymphatic organs within a few days. Around the 10th day, detecting the virus in the blood is possible. Next, the virus spreads exponentially over several weeks, peaking around day 30 [26]. Then, antibodies against HIV become detectable. The more virus particles in the bloodstream, the more host cells are attacked, resulting in a decrease in the number of CD4^+^ T cells. The increase in viral load is associated with a gradual decrease in the CD4^+^ cell count. As a result, the organism’s immune response weakens usually over several years. The CD4^+^ cell count applies in monitoring the course of the disease. The decrease of CD4^+^ cell count below 200 cells/mm^3^ results in severe dysfunction of the immune system, and then a diagnosis of AIDS is made. At this stage of the disease, untreated people usually develop opportunistic infections, such as toxoplasmosis [30,31]. Kodym et al. [32] suggested that the reactivation of toxoplasmosis can occur in HIV and *T. gondii* co-infected subjects with not only the lowest CD4^+^ count but also with decreased number of CD8^+^ and NK cells. These results are consistent with recent research reporting the pivotal role of CD8^+^ in controlling parasite reactivation during HIV infection [33,34]. CD4^+^ cells provide essential help with the induction of the primary cell response of CD8^+^ cells. They help CD8^+^ cells mainly by facilitating the presentation of antigens and up-regulation of costimulatory molecules on dendritic cells to levels optimal for the induction of a robust CD8^+^ cell response [34]. Since both CD4^+^ and CD8^+^ cells are essential in the humoral response to the presence of the parasite, the depletion of both cell types results in increased susceptibility to toxoplasmosis. This is particularly important in the late phase of HIV infection when patients are most deficient in CD8^+^ cells [34].

The original TORCH complex describes congenital infections caused by *Toxoplasma gondii*, Rubella virus, Cytomegalovirus (CMV), and the Herpes Simplex Virus types 1 and 2 (HSV-1, HSV-2). In the case of HIV-positive patients, the pathogens mentioned above can also cause severe, life-threatening infections. Some of them are preventable, and some can be cured to some extent. Singh et al. [35] indicated that monitoring HIV-positive patients for these infections is extremely important. They emphasized that TORCH follow-up in HIV/AIDS cases can reduce overall morbidity and mortality. Thus, the immune profile of TORCH in HIV-infected patients has an excellent prognostic significance [35]. In immunocompromised hosts, the latent toxoplasmic infection may be reactivated due to the conversion of bradyzoites to tachyzoites. Infection in this group of patients can cause severe disability or be life-threatening [36]. It should be emphasized that the infection can take place despite the history of toxoplasmosis in the past and the presence of IgG antibodies. This makes immunocompromised patients vulnerable to developing the disease anyway. It has been indicated that regular monitoring (and treatment if indicated) is necessary for these individuals. It is also reasonable to use other diagnostic methods because monitoring changes in antibody titer gives a delayed picture of disease activity.

Toxoplasmosis in HIV-positive patients is usually a consequence of the reactivation of parasite tissue cysts, leading to TE, headache, confusion, seizure, lethargy, and focal neurological symptoms [37]. TE is the most frequent clinical manifestation of toxoplasmosis in HIV-infected people [30]. In patients coinfected with HIV and toxoplasmosis, the risk of TE ranges from 30 to 40% [38]. Some TE patients can also exhibit the symptoms of neuropsychiatric disorders, including dementia, anxiety, psychosis, and personality disorders [21]. Moreover, in HIV-positive patients, toxoplasmosis can increase susceptibility to opportunistic infections by various mechanisms, including the depletion of CD4^+^ lymphocytes, decreased production of type 1 cytokines, as well as reduced activity of cytotoxic lymphocytes [39]. Moreover, cases of disseminated toxoplasmosis with fever, disseminated intravascular coagulation, sepsis-like syndrome with hypotension, and elevated lactate dehydrogenase and pulmonary dehydrogenase may occur [21].

Wu et al. [30] emphasized that poorer cognitive function may accompany toxoplasma seropositivity. They noticed mild to moderate neurocognitive impairment in nearly half of HIV-infected people. They suggested that coinfection with HIV and toxoplasma should be considered in the differential diagnosis of young patients with rapidly progressive memory loss. HIV-infected individuals with peripheral blood CD4^+^ T cell count lower than 200/μL, who are toxoplasma IgG-seropositive, and do not receive trimethoprim-sulfamethoxazole treatment are at particular risk of TE [30,31]. TE generally follows the reactivation of latent infection [30]. Administration of antiretroviral therapy can reduce the risk of developing TE two-fold. Thus, primary antiretroviral treatment is recommended for HIV-infected *Toxoplasma* IgG-seropositive patients with peripheral blood CD4^+^ T cell levels below 100/μL [31]. TE can be diagnosed in AIDS patients by examination of cerebrospinal fluid (CSF) or serum anti-*Toxoplasma* IgG and/or IgM extended by imaging examinations of brain structures for the presence of lesions in magnetic resonance imaging (MRI) [30]. However, it should be taken under consideration that the parasite is present in CSF only intermittently, so a negative CSF result does not exclude TE and should be confirmed by repeated tests [30].

## 4. Diagnostics of Toxoplasmosis

The detection of *T. gondii* in fecal, tissue, water, and environmental samples has traditionally been accomplished by microscope examination. However, serological and molecular methods are more sensitive and reliable than identification based on light microscopy [40]. Table 1 presents serological methods used in the diagnosis of toxoplasmosis in humans [22,41].

Immunological methods are based on detecting parasite surface antigens by host-specific immunoglobulins. Serological methods were preferentially used for diagnostics of toxoplasmosis in past decades. However, in addition to undoubted advantages, these tests have significant limitations [22,41]. The Sabin–Feldman dye test (SFDT) is a “gold standard” for the serological detection of anti-*Toxoplasma* IgG and IgM antibodies. The test explores changes in the cytoplasm of living toxoplasma trophozoites caused by the action of antibodies present in the blood serum of the infected individual; these changes become clearly visible after staining the toxoplasma with a strong solution of methyl blue [40,53]. However, the use of live tachyzoites is required in this method, which is not possible in most laboratories [54]. Therefore, an enzyme-linked immunosorbent assay (ELISA) remains the most common method used in clinical laboratories [22,41]. Chemiluminescence assay (CLIA) and enzyme-linked fluorescence assay (ELFA) are variations of the standard ELISA. These methods have several benefits, including cost-effectiveness and fast and precise measurement of antibody levels. Furthermore, CLIA is especially useful in determining the IgG avidity index [22]. Moreover, indirect fluorescent antibody test (IFAT), indirect hemagglutination assay (IHA), immunosorbent agglutination assay (ISAGA), and modified agglutination test (MAT) have also been used [55]. Serological diagnosis of toxoplasmosis takes advantage of the persistent presence of specific antibodies in serum following exposure to the parasite [41]. Toxoplasmosis is primarily diagnosed by serological detection of IgM and IgG antibodies, and sometimes IgA directed against parasitic protein antigens (see Figure 1) [41,55]. Upon primary exposure to the parasite, in response to the parasite’s presence, antibodies in the M class are first formed. Thus, the presence of IgM antibodies may indicate a recently acquired, acute infection [55]. Subsequently, IgG immunoglobulins are formed, and they persist for years [41]. Because IgM antibodies sometimes persist in the blood serum for months or years after the initial infection, the use of IgG avidity assay has enabled a better estimation of the time of infection acquisition and identification of the primary infection with *T. gondii* [49]. Avidity is the aggregate strength by which a mixture of polyclonal IgG molecules reacts with multiple epitopes of the proteins. The IgG avidity increases progressively after immunities from infections and is referred to as maturation of the humoral immune responses. Low IgG avidity indices are usually characteristic of the first few months of primary infections, while high-avidity indices specify nonprimary infections [49]. The ELISA method can use both native antigens and recombinant and chimeric ones. In the recently developed approach to improving the diagnosis of *T. gondii* infection, the ELISA method increasingly relies on recombinant antigens instead of native antigens [41].

An alternative method for the diagnosis of toxoplasma infection is to examine circulating antigens. Surface antigen 1 (SAG1) is the most explored and used tachyzoite stage-specific antigen of *T. gondii* for serological test development (see Figure 1) [43]. SAG1 is a significant parasite surface protein which plays a role as a critical virulence factor. This protein induces both humoral and cellular host immune responses [14]. DELAA is one of the novel methods developed to detect SAG1 [43]. This method uses aptamers, termed “chemical antibodies”, which characterize high binding affinity and specificity [45]. Additionally, a real-time polymerase chain reaction (real-time PCR) assay using a set of primers targeting the SAG1 gene showed a high sensitivity for the fast and specific detection of parasites [43].

Bead-based multiplex assays (BBMAs), which use color-coded beads that carry the antigen of interest, are high throughput methods for the simultaneous detection and quantification of multiple analytes in a sample [56]. Garg et al. [55] used a BBMA containing a synthetic phosphoglycan portion of the *Toxoplasma gondii* glycosylphosphatidylinositol 1 (GPI1) to detect GPI1-specific antibodies in human sera. The glycan was conjugated to beads at the lipid site to retain its natural orientation and its immunogenic groups. The authors demonstrated that GPI1 was a more reliable predictor for parasite-specific IgM response than SAG1, indicating that BBMA using GPI1 in combination with SAG1 may strengthen *Toxoplasma gondii* serology diagnostics.

A wide range of DNA-based detection assays has been developed for the parasite detection. Molecular biology methods allow the detection of the parasite DNA in various biological samples, including whole blood, CSF, urine, bronchoalveolar fluid, amniotic fluid, aqueous humor, cord blood, placental and brain tissues, and other samples where the parasite may be present [22,40]. Amplification of *T. gondii* DNA by quantitative polymerase chain reaction (qPCR) has been performed in different biological materials, including blood and tissue samples, amniotic fluid, and CSF. The B1 gene (35 copies) and the 529-bp repeat element (REP 529, more than 300 copies) have been extensively explored as targets for PCR detection [57]. The rep529-PCR has a higher sensitivity than B1-PCR (100% vs. 82.7–90%, respectively) due to higher repeat numbers in the parasite genome [58]. Nested PCR utilizing the recombinant dense granule antigen 7 (GRA7) gene has been recommended as an excellent method to detect *T. gondii* infection. GRA7 has an important function in the parasitophorous vacuole. It is involved in the sequestration of host endolysosomes and is expressed in all infectious stages, including bradyzoite, tachyzoite, merozoite, and sporozoite. Amplification of conserved regions in the GRA7 gene is characterized by specificity and sensitivity similar to the 529-bp fragment and is more sensitive than the B1 gene [33].

An isothermal nucleic acid amplification technology (INAAT) may also be useful in the diagnostics of toxoplasmosis. It allows DNA amplification to be performed rapidly. Two INAAT methods, namely Loop-Mediated Isothermal Amplification (LAMP) and Nucleic Acid Sequence-Based Amplification (NASBA), have been used for the rapid detection of *Toxoplasma gondii* [33,59]. LAMP is based upon the amplification of DNA (or RNA if combined with a retro-transcription step) under isothermal conditions (63–65 °C) [43]. This method, initially developed by Notomi et al. in 2000 [60], is based on the reaction of DNA polymerase with strand displacement activity and a set of two specially designed inner primers (forward inner primer, FIP, and backward inner primer, BIP) and two outer primers (forward primer F3 and backward primer B3) that amplifies DNA rapidly with high efficiency and specificity. The LAMP method can amplify a few copies of genetic material to 109 copies in less than an hour [61]. The advantage of this technique is that DNA amplification can be easily detected by visual inspection of the turbidity or fluorescence of the reaction mixture or by a Loopamp real-time turbidimeter [62]. During the reaction, pyrophosphate ions released from deoxyribonucleotide triphosphates (dNTPs) as by-products react with magnesium ions, forming a white precipitate of magnesium pyrophosphate. As a result, a positive reaction can be visually inspected immediately or by observing the color change after the post-addition of DNA intercalating fluorescent dye SYBR green I or pre-addition of the colorimetric indicator hydroxynaphthol blue (HNB) to the reaction mixture. The color change is permanent, and thus, it can be kept for record purposes. The use of gel electrophoresis in this technique is not mandatory, and thus the risk of carry-over contamination is reduced [61]. Fallahi et al. [62] determined that the LAMP assay targeting the RE and B1 genes had superior sensitivity and specificity than the conventional PCR technique based on the same genomic targets for the detection of the *T. gondii* DNA from both mice-obtained tachyzoites and clinical samples of leukemic patients. The NASBA method involves the simultaneous use of three enzymes, namely avian myeloblastosis virus reverse transcriptase (AMV-RT), RNase H, and T7 RNA polymerase, under isothermal conditions (41 °C). The final amplification product includes a single-stranded RNA of the opposite sense of the original target [63]. What is important is that NASBA does not require a thermocycler. A constant temperature throughout the amplification reaction enables each reaction step to proceed when intermediate amplification becomes available, so the NASBA reaction is more efficient than DNA methods that are limited to binary increases per cycle [63].

Because many pathogens that threaten human health are foodborne parasites, in addition to the diagnostic methods mentioned above, there is also a need to test food for contamination by parasites. Detection of *T. gondii* infection in animals or animal products has been reported in numerous publications [63,64,65,66,67,68,69,70,71]. It has been indicated that, in the USA, adult sheep and lambs had high *T. gondii* seroprevalence, and thus they should be considered a significant source of the parasite for humans [72]. Temesgen et al. [73] developed and evaluated novel multiplex qPCR for simultaneous detection of *T. gondii*, *Echinococcus multilocularis*, and *Cyclospora cayetanensis* in berry fruits. The proposed method is characterized by high specificity and precision, with the detection limit of *T. gondii* equal to 10 oocysts in 30 g of berries.

## 5. Diagnostics of Toxoplasmosis in HIV-Infected Patients

Cerebral toxoplasmosis is one of the most common causes of focal brain damage in HIV patients. MRI plays an essential role in diagnosing and treating cerebral toxoplasmosis [74,75]. Imaging techniques cannot distinguish toxoplasmosis from other neurological opportunistic infections. Thus, early laboratory diagnosis and prompt treatment of cerebral toxoplasmosis in HIV patients could reduce the risk of neurological damage and mortality [36].

The serological diagnosis is the first and the most frequently used approach to determining the stage of infection. The detection of IgG, IgM, IgA, and IgG avidity by different methods offers this opportunity [76]. The IgG avidity test is a qualitative test that differentiates chronic toxoplasmosis caused by a recent infection and determines the status of toxoplasma infection. High IgG avidity indicates chronic or reactivated infection, while low IgG avidity suggests acute infection [77]. Lui et al. [78] noticed that AIDS/HIV patients have higher seroprevalence of *T. gondii* than HIV-negative individuals. Smoguła et al. [4] also observed higher seroprevalence of *T. gondii* in subjects at increased risk of HIV infection. Singh et al. [35] described higher seroprevalence (21.94%) of IgM antibodies and lower (11.11%) IgG antibodies against *T. gondii* among HIV/AIDS patients in India. However, Zhou et al. [79] pointed out that the reactivation of latent toxoplasmosis is more common than primary infection in patients with HIV. Nourollahpour Shiadeh et al. [80] analyzed the prevalence of acute (AT) and latent toxoplasmosis (LT) in HIV-infected pregnant women. They estimated AT in 1.1% and LT in 45.7% of pregnant women. It should be considered that, in HIV-infected patients, the cellular response is totally altered. Due to the depletion of T-helper lymphocytes (CD4^+^ cells), reduced activity of CD8^+^ cytotoxic T cells, and decreased production of type 1 cytokines, HIV-infected patients are at greater risk of developing opportunistic infections [80]. Th1/Th2 imbalance and a decrease in CD4^+^ T cells (CD4^+^ T-lymphocyte counts below 100 cells/µL) can cause the reactivation of latent toxoplasmosis in HIV patients due to the failure of antiparasitic CD4^+^ T-cell response [36].

It has been proven that *T. gondii* can be transmitted through the transfusion of whole blood or leukocytes. The parasite could stay alive in the citrated blood sample at 5 °C for more than 50 days. Thus, the refrigeration of blood specimens during storage cannot prevent toxoplasma transmission by transfusion [38]. Toxoplasmosis could be a lethal opportunistic infection after allogeneic hematopoietic cell transplantation [58]. Therefore, using both serologic and molecular methods in blood donors would be highly important for safe blood donations [81]. To prevent toxoplasmosis, serological screening of donors and recipients before transplantation is of primary importance to identify patients at high risk of toxoplasmosis, i.e., seropositive hematopoietic stem cell transplant recipients and mismatched (seropositive donor/seronegative recipients) solid organ transplant recipients. Preventing toxoplasmosis disease in those patients presently relies on prophylaxis via the prescription of co-trimoxazole [82]. Foroutan et al. [83] analyzed studies on *T. gondii* seroprevalence in hemodialysis patients. They observed specific IgG antibodies in 58% and specific IgM antibodies in 2% of hemodialysis patients, respectively, and 40% and 0% in controls. Furthermore, the authors suggested that hemodialysis patients, as well as HIV patients, should be screened periodically for *T. gondii* infections due to their reduced immunity.

The antibody-based methods are not always appropriate for differentiation between acute and reactivated toxoplasmosis. High antibody titers may indicate active disease (i.e., positive IgM or low avidity of IgG) or may point to a higher risk of developing the disease (permanently positive IgG) [43]. Routine anti-*T. gondii* IgM diagnostic tests show limited value in diagnosing HIV patients because they have limitations such as their inefficiency for confirmation of the parasite infection in immunocompromised subjects [22,77]. The IgG avidity test is a confirmatory test for differentiating between acute and chronic phases of the infection but is not specific enough in immunocompromised patients [84]. Serological tests may fail to detect the disease in immunocompromised patients because the titers of anti-*Toxoplasma* antibodies may fail to rise [63]. Thus, it is recommendable to detect parasite DNA in patients with advanced HIV disease [81]. It should be emphasized that molecular diagnostics do not depend on the immunological condition of the host. Thus, molecular methods would be the best option for immunocompromised patients [83]. The direct demonstration of the presence of parasites in tissues or body fluids would be a breakthrough for the diagnosis of toxoplasmosis. Therefore, molecular methods are considered important diagnostic tools for detecting toxoplasma infection in immunosuppressed hosts [77]. 

Nowadays, the use of the PCR method has been helpful in the improvement of diagnostic procedures, especially acute toxoplasmosis diagnosis [39]. Real-time PCR methods can detect fewer copies of the parasite genome compared to conventional PCR methods [85]. Currently, in Poland, according to the recommendations of the Polish AIDS Society, all symptomatic patients should be tested with PCR for the presence of parasite DNA in CSF [86]. Primary anti-parasitic treatment of toxoplasmosis in HIV patients should be continued for more than 6 weeks. Moreover, the maintenance treatment should be continued until the CD4^+^ T cell count reaches more than 200 cells/μL in two tests repeated 6 months apart and after the symptoms of the acute disease subside [86]. These recommendations are generally in line with “Guidelines for the Prevention and Treatment of Opportunistic Infections in Adults and Adolescents with HIV”, available at https://clinicalinfo.hiv.gov/en (accessed on 27 November 2022) [87]. The qPCR assay is currently the state-of-the-art molecular technique in toxoplasmosis diagnostics in HIV patients. This technique allows rapid parasite DNA detection with little risk of laboratory contamination with amplicons [88]. There are also promising studies indicating the potential use of micro-RNA (miRNA) and circulating RNA (circRNA), especially miR-21-5p and miR-146a-5p molecules, in the diagnosis of HIV/cerebral toxoplasmosis co-infections. Still, their implementation as diagnostic targets requires further research [89,90]. 

In our opinion, there is a need to constantly search for new, effective, high-sensitive, and high-specific diagnostic methods that detect the parasite’s infection. Therefore, molecular biology methods have great diagnostic potential. Although most molecular methods generate high costs, the LAMP method seems to be an interesting alternative. This method is relatively inexpensive and fast and, at the same time, based on a portable and cost-effective device. In recent years, LAMP has been applied in the diagnosis of other parasitic diseases, such as malaria, trypanosomiasis, schistosomiasis, or leishmaniasis [91,92,93,94].

## 6. Conclusions

In conclusion, the different serology and molecular methods used in detecting toxoplasma infection have their advantages and limitations. It is crucial to use appropriate, combined diagnostic methods depending on the needs of specific clinical cases. Timely and appropriate diagnosis is extremely important for the health of people infected with HIV, especially regarding the risk of toxoplasmic encephalitis. As HIV-positive people, despite a history of toxoplasmosis, are still at risk of developing an acute course of the disease, regular monitoring for potential reactivation of the infection is necessary. Therefore, it is essential to use modern, sensitive, and specific methods that allow proper monitoring of the course of the disease. Traditional diagnostics based on antibody titer determination may be inappropriate due to the delayed reaction of antibody formation. Undoubtedly, modern techniques of molecular biology should be recommended to improve the diagnosis of toxoplasma, particularly in HIV patients. 

## Figures and Tables

**Figure 1 pathogens-12-00110-f001:**
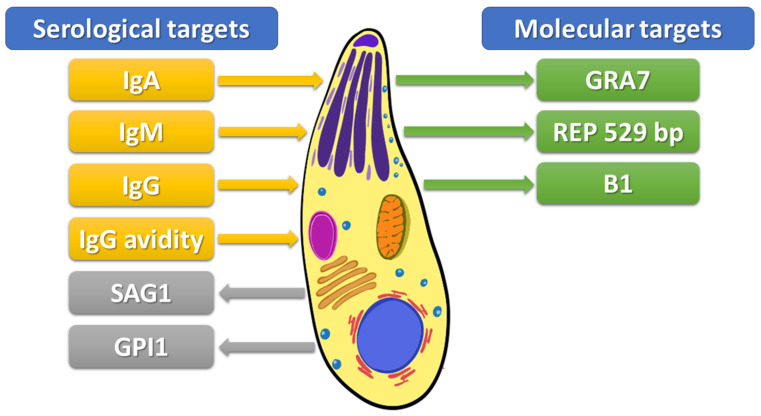
Targets to detect *Toxoplasma gondii* infection. IgA–Immunoglobulin A, IgM–Immunoglobulin M, IgG–Immunoglobulin G, IgG avidity–avidity of immunoglobulin G, SAG1–surface antigen 1, GPI1-glycosylphosphatidylinositol 1 antibody, GRA7-recombinant dense granule antigen 7 gene, REP 529-529-bp repeat element, B1–35-fold repetitive B1 gene.

**Table 1 pathogens-12-00110-t001:** Serological methods used in the diagnostics of *Toxoplasma gondii* infection.

Abbreviation	Test Name	Refs
CLIA	chemiluminescence assay	[22,42]
DELAA	direct enzyme-linked aptamer assay	[43,44]
ELFA	enzyme-linked fluorescence assay	[22,45]
ELISA	enzyme-linked immunosorbent assay	[41,46]
ICT	immunochromatographic test	[40,47]
IFAT	indirect fluorescent antibody test	[40,48]
IgG avidity	immunoglobulin G (IgG) avidity test	[40,49]
IHA	indirect hemagglutination test	[40,50]
ISAGA	immunosorbent agglutination assay	[40,51]
LAT	latex agglutination test	[40,52]
MAT	modified agglutination test	[48,50]
SFDT	Sabin Feldman dye test	[40,53]
WB	western blotting	[46,53]

## Data Availability

The data presented in this study are available on request from the corresponding author. The data are not publicly available due to privacy or ethical restrictions.

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
