# Peer review of "Advances and Challenges in Diagnostics of Toxoplasmosis in HIV-Infected Patients"

_pathogens, 2023, doi:10.3390/pathogens12010110_

Round 1
Reviewer 1 Report
The authors have reviewed on the challenges in the diagnostics of toxoplasmosis facing HIV-infected patients. This is an important issue that has continuously been faced by this group of toxoplasmosis patients and this review will certainly raise awareness.
However, the manuscript can still be further improved if the authors have brought up the point as to why toxoplasmosis is a serious infection and present huge complications for this group. In particular, the review will stand out even more if the authors can comment on the potential of toxoplasmosis as a sexually transmissible infection. This has been witnessed in animals such as dogs and goats. And more recently and interestingly, this has been demonstrated in humans as well: Presence of Toxoplasma gondii tissue cysts in human semen: Toxoplasmosis as a potential sexually transmissible infection (https://doi.org/10.1016/j.jinf.2022.10.034). I think commenting and adding references for these papers in the opening introduction will allow authors to appreciate and fully understand the real concern of toxoplasmosis for HIV-infected patients. Other than this point, the review is well-thought out and ready for acceptance when the above comment is addressed.
Reviewer 2 Report
The manuscript from Roland et al aims to detail "Advances and challenges in diagnostic" approaches to Toxoplasma gondii. Their information is generally accurate, up-to-date, and will be potentially informative to a non-clinical reader who is very familiar with immunological assays. Recommended revisions largely address increasing readership audiences, provide clarity, and leveraging expertise garnered from the review to more strongly state or provide conclusions.
-Table 1 and the associated text details a number of serological assays, which generally are not explained in the same detail as LAMP and other PCR-based detection methods. It seems reasonable that these immunologic approaches also receive descriptive information to allow the reader to understand these approaches, particularly those less familiar with them. Also, consider adding at least one citation to each approach.
-Figure 1 could be improved. The left column of serological targets could separate the Igs from SAG1 and GPI1 with color or some other indicator beyond the nondescript arrow direction.
-Throughout the manuscript, and particularly in relation to molecular target detection, the fluid/tissue/biomatter source is not listed, e.g. blood or CSF. Including this information will help guide clinicians and others potentially developing next-gen diagnostics.
-The final paragraph before Conclusions includes present utilization in Poland about practice in relation to HIV-positive patients. This section, or the Conclusion section, should include more recommendations from the authors about the validity of those approaches, and/or their recommendations of which approaches (specifically) might be advisable in the clinic. Furthermore, future diagnostic approaches in Toxo could be expanded, as well as advanced/next-gen diagnostics being applied in other diseases/organisms that may be beneficial in Toxoplasma.
-The introductory paragraph is missing English articles (a, an, the) in a few places.
Reviewer 3 Report
In the review entitled “Advances and challenges in diagnostics of toxoplasmosis in HIV-infected patients” the authors discuss the relevant issues associated with toxoplasmosis in HIV-infected immunodeficient patients. Below are the comments.
1. Lines# 70-71: “Therefore, they highlighted the importance of routine testing for T. gondii infection in all HIV-infected people.” The sentence has been corrected. The structure of sentences should be checked throughout the manuscript thoroughly.
2. Line# 107: “It primarily attacks CD4+ T cells [23,25]”. The cell phenotype markers (CD) positivity symbols should be superscripts. Whenever cell phenotypes are described, CD notation with positivity should be used consistently.
3. Lines# 356-359: “There are also promising studies indicating the potential use of micro-RNA (miRNA) and circulating RNA (circRNA), 357 especially miR-21-5p and miR-146a-5p molecules, in the diagnosis of HIV/cerebral toxoplasmosis co-infections, but their implementation as diagnostic targets requires further research [32,74,75].”
4. The authors are focusing mostly on the detection of either parasite antigens or the humoral compartment of the host. However, in addition to these aspects, the cell-mediated immune compartment of the host can also be tapped for the diagnostics. For example, Khan and co-workers have investigated on this aspect and reported the exhaustion status of CD4+ T-cells, leading to CD8+ T-cell dysfunction and reactivation of toxoplasma in chronically infected patients (Khan et al., 2019). The authors should discuss this aspect in detail.
[Khan IA, Hwang S, Moretto M. Toxoplasma gondii: CD8 T Cells Cry for CD4 Help. Front Cell Infect Microbiol. 2019 May 1;9:136. doi: 10.3389/fcimb.2019.00136. PMID: 31119107; PMCID: PMC6504686.]
Round 2
Reviewer 1 Report
The authors have addressed my comments.
Author Response
We would like to thank the Reviewer for the careful review of our manuscript.
Reviewer 2 Report
All requests by reviewers have been achieved, and the manuscript would likely provide a value to the field. Only minor spelling and capitalization errors exist.